# Synthesis of Imidazole-Based Medicinal Molecules Utilizing the van Leusen Imidazole Synthesis

**DOI:** 10.3390/ph13030037

**Published:** 2020-03-03

**Authors:** Xunan Zheng, Zhengning Ma, Dawei Zhang

**Affiliations:** 1College of Chemistry, Jilin University, Changchun 130012, China; zhengxn8217@mails.jlu.edu.cn (X.Z.); 2019203060028@whu.edu.cn (Z.M.); 2College of Plant Science, Jilin University, Changchun 130062, China; 3Key Laboratory of Combinatorial Biosynthesis and Drug Discovery (Ministry of Education), School of Pharmaceutical Sciences, Wuhan University, Wuhan 430071, China

**Keywords:** van Leusen, TosMICs, imidazole, synthesis

## Abstract

Imidazole and its derivatives are one of the most vital and universal heterocycles in medicinal chemistry. Owing to their special structural features, these compounds exhibit a widespread spectrum of significant pharmacological or biological activities, and are widely researched and applied by pharmaceutical companies for drug discovery. The van Leusen reaction based on tosylmethylisocyanides (TosMICs) is one of the most appropriate strategies to synthetize imidazole-based medicinal molecules, which has been increasingly developed on account of its advantages. In this review, we summarize the recent developments of the chemical synthesis and bioactivity of imidazole-containing medicinal small molecules, utilizing the van Leusen imidazole synthesis from 1977.

## 1. Introduction

Imidazole ring, which is widely found in natural products and medical molecules, is one of the most prominent, five-membered, nitrogen-containing, heterocyclic scaffolds. Furthermore, imidazole-based heterocyclic compounds, which possess a vital position in medicinal chemistry, have been playing a central role in the treatment of numerous types of diseases, and new derivatives for medicinal use are being energetically developed worldwide [1,2,3,4,5]. Due to the peculiar structural characteristic of imidazole scaffold with a worthy electron-rich feature, it is advantageous for imidazole groups to combine with various receptors and enzymes in biological systems, through diverse weak interactions, thereby showing a variety of biological activities. At present, a legion of imidazole-containing compounds with high a medical potential as a clinical drug have been widely used to treat diverse types of illnesses, such as antibacterial [6,7], antifungal [8,9], anti-inflammatory [10,11], antiviral [12,13], anti-parasitic [14,15], anticancer [16,17], antihistaminic [18,19], and enzyme inhibition [20,21]. Imidazole and its derivatives encompass a vast range of medical activities, as shown in the following Table 1.

Owing to the significant pharmacological or biological activities and the enormous medicinal value of imidazole-based molecules, the synthesis of the imidazole-skeleton small molecule has been paid attention to by pharmaceutical chemists and organic synthesis researchers. However, there is still a need for a simple and efficient way to construct the imidazole heterocyclic skeleton. In recent decades, there have been numerous classical strategies for synthesizing this ring compound in the laboratory, including van Leusen imidazole synthesis [22], Debus-Radziszewski imidazole synthesis [23], Wallach imidazole synthesis [24], etc. Among these synthetic strategies, it is well-known that the van Leusen imidazole synthesis based on TosMICs, which is the cycloaddition reaction, is one of the most convenient and attractive protocols for the preparation of imidazole-based small molecules, due to its excellent advantages like simple manipulation, easily obtained raw materials and a wide range of substrates, which has been developed rapidly in the past decades (Scheme 1). 

TosMIC, one of the most significant reactants, has many good features at room temperature, including being a stable solid and being odorless and colorless. Since it was introduced and applied in organic synthesis by the Dutch professor van Leusen in 1972, this reagent is also known as van Leusen’s reagent. Up to now, TosMIC and its derivatives have been recognized as one of the most significant building blocks in nitrogen heterocyclic synthesis, which have been fruitfully employed especially in the preparation of imidazole-based heterocycles [25,26,27,28,29]. 

Therefore, this review will summarize the developments of the synthesis of imidazole-based molecules, utilizing the van Leusen imidazole synthesis, based on TosMICs from 1977. It is expected that this review article will be beneficial for new opportunities to search for a reasonable design for less toxic and higher bioactive imidazole-containing drugs.

## 2. General van Leusen Imidazole Synthesis

In 1977, van Leusen et al. first published that TosMIC and aldimine undergo a base-induced cycloaddition reaction in a proton solvent, meanwhile, the effects of R^1^ and R^2^ on the formation of **17** were qualitatively analyzed. They discovered that *α*-tosylbenzyl isocyanate and *α*-tosylethyl isocyanate **18** could form 1,4,5-trisubstituted imidazoles **19** (Scheme 2). Based on the various advantages of this reaction, it is widely named as the van Leusen imidazole synthesis [22].

As shown in Scheme 3, the van Leusen imidazole synthesis allows the preparation of imidazole through [3 + 2] cycloaddition reaction from aldimines and a reaction with TosMICs, which contain reactive isocyanide carbons, active methylene, and leaving groups such as C2N1 “3-atom synthon”. The cyano moiety can be a gradual cycloaddition to polarize a double bond under a base condition. The elimination of *p*-TosOH forms the intermediate 4-tosyl-2-imidazoline to produce the target 1,4,5-trisubstituted imidazoles **19**, accompanied by the elimination of *p*-TosOH, which is negative to the obtained 1,5-disubstituted imidazoles.

## 3. Developments of the van Leusen Imidazole Synthesis

In 1995, Frankowski and co-workers were reported to lead the known *D*-xylo-pentodialdose to form imidazo-*L*-xylo-piperidinose derivatives, based on a sequential eight-step reaction. In this process, the imidazole-base molecule was obtained through a van Leusen reaction as a key step. As shown in Scheme 4, applying this van Leusen methodology, **20** and TosMIC **18** were transformed into the imidazole derivative **21**. At last, removal of the protecting group gave to the target product **22**, which is a kind of bicyclic azasugar and a glycosidase inhibitor [30]. 

In 1998, the Sisko group synthesized the 1,4,5-trisubstituted imidazole **25**, which displayed potent binding with p38 MAP kinase, a recently discovered protein kinase that participates in an inflammation regulatory mechanism. As shown in Scheme 5, 1,4,5-trisubstituted imidazole **24** was accessed by a novel and facile protocol, based on the reaction of an *α*-ketoaldimine **23** with an aryl-substituted TosMIC reagents **18** as the key step [31]. 

In 2000, they also described an active and gentle procedure for preparing multisubstituted imidazoles in one-pot, from an aryl-substituted TosMIC and a generated imine, in situ. One of the reactions is shown in Scheme 6, cycloaddition of imine **28**, prepared imine in situ from a 40% aqueous solution of pyruvaldehyde **26** and amine **27**, then, ketone **29** was prepared in DMF, using aryl-substituted TosMIC **18** and K_2_CO_3_, with a 75% yield. Surprisingly, when the reaction was carried out at 0 °C, the product **29** was blended with 15–20% of the 1,4-disubstituted imidazole **30** [32].

In the same year, the Vanelle group converted 6-nitropiperonal into the methylimine **31** through a condensation reaction with methylamine in ethanol. The molecule with imidazole ring **32** was formed by reacting equimolar quantities of TosMIC **18** and aldimine **31** with K_2_CO_3_ in methanol solvent, under a refluxing condition. Then, through the classical Knoevenagel reaction, compound **33** were obtained, which displayed an effective in vitro leishmanicidal bioactivity (Scheme 7). Compound **33a**, higher in vitro leishmanicidal bioactivity, might lead to a promising significant therapeutic agent (Scheme 8) [33].

From 2005 to 2006, in order to access unusual imidazole-based heterocyclic structures, a series of experimental results were reported by the Gracias and Djuric group.

Initially, they reported a new protocol employing the van Leusen three-component reaction and the ring-closing metathetical reaction in a sequence method, to form a fused bicyclic imidazole ring. The general method is shown in Scheme 9 (top), the reaction underwent smoothly with the condensation reaction of 4-pentenal **34** and allylamine **35**, in DMF at room temperature, to give the imine in situ, which was followed by the addition of the TosMIC reagent **18** and a base, to access the van Leusen imidazole product **36** in high yield. Next, a ring-closing metathesis (RCM) reaction was performed through the second-generation Grubbs catalyst to get imidazole **37** and their derivatives (Scheme 9, top) [34].

Next, they developed a concise route of fused imidazo azepine analogs via the stepwise van Leusen/intramolecular enyne metathesis synthesis. The condensation reaction of 4-pentenal **34** with but-2-yn-1-amine **38** in DMF at room temperature gives the imine in situ, and then adds phenyl TosMIC **18** and K_2_CO_3_ to access the van Leusen imidazole product **39** in high yield. Subsequent cyclization via the intramolecular enyne metathesis reaction results in the formation of the cyclized product **40** containing the diene functional group and their derivatives (Scheme 9, bottom) [35]. 

In late 2005, they also reported a facile route of fused triazolo imidazole derivatives via a sequential van Leusen/alkyne–azide cycloaddition reaction. The use of an azide functionality on the aldehyde **41** and an alkyne functionality on the amine **42** gives the bifunctional raw material for the van Leusen reaction resulting in substrate **43**. Subsequent cyclization through the intramolecular alkyne–azide cycloaddition will form the fused triazolo imidazole skeletons **44** (Scheme 10) [36].

Then in 2006, they continued to publish the methods for the fused imidazole ring synthesis by using the tandem van Leusen/RCM, van Leusen/enyne metathesis, van Leusen/alkyne–azide cycloaddition, or van Leusen/Heck reaction, respectively.

Firstly, they synthesized fused imidazo-pyridine and imidazo-azepine derivatives by using a sequential van Leusen/ intramolecular Heck route. The imidazole **47** and **49** were accessed via the condensation reaction between an appropriate aldehyde-containing vinylogous bromide **45** and an amine-containing double bond **46,** respectively. Then, preformation of the imine, the desired TosMIC reagent **18** and K_2_CO_3_ were added, and the cyclization was allowed to proceed at room temperature to get imidazo-[1,5-*a*]pyridine **48** or imidazo[1 ,5-*a*]azepine **50** (Scheme 11) [37].

Next, they described an elegant method to obtain a fused imidazole ring, employing the van Leusen three-component reaction, followed by Pd/Cu catalyzed intramolecular *C*-arylation. Meanwhile, another available protocol to form 5,6-dihydroimidazo[2,1-a]-isoquinoline system **53**, via a radical cyclization reaction of tethered aryl halides onto the imidazole ring, was also published. In their article, the imidazole precursors were accessed through an alkylation chemistry, and the van Leusen reaction was used to assemble imidazole followed by a transition metal catalyzed C–C bond framework (Scheme 12) [38].

In 2006, Dömling’s group reported that substituting pyrroloimidazoles **56** were assembled by van Leusen multicomponent reaction (MCR) of TosMIC **18**, indole carbaldehydes **54**, and primary amines **55**. They were interested in bioactivity of the products and these imidazole-containing skeletons were screened in a phenotypic assay for neurite outgrowth. The test results indicated that these small molecules would serve as useful chemical probes to research neurite growth and might be used as a therapeutic application for axon regeneration of lesions of the human spinal cord and brain (Scheme 13) [39].

Based on their previous research, they also reported a novel synthesis route of potential aspartyl-protease inhibitors in 2007. In their method, 1,4,5-trisubstituted 1-(4-piperidyl)-imidazoles **59** could be obtained through an isocyanide-based van Leusen MCR of *α*-substituted TosMICs **18**, aldehydes **57**, and 4-aminopiperidine **58** (Scheme 14) [40].

In 2009, a range of 1,5-disubstituted-4-methylimidazole preparations were reported by Fodili and co-workers. Compounds **63** were prepared from acetimine analogs **62**, which were obtained from dehydroacetic acid **60** and primary amine **61**. These were enabled to react with TosMIC **18** in the presence of catalytic amounts of bismuth triflate. A plausible mechanism was put forward that involved the first formation of an imidazoline intermediate, followed by methyl migration, and then subsequent aromatization provided access to the target imidazole ring (Scheme 15) [41].

In 2012, Hulme and Moliner reported a novel synthesis of two pharmacological relevant classes of molecules forming the imidazoquinox aline scafflod **66** and **68**, respectively. This method involves the use of 1,2-phenylenediamines **64** and glyoxylic acid derivatives, namely ethyl glyoxylate **65** or benzylglyoxamide **67**, along with TosMIC **18** in a microwave-assisted van Leusen three-component condition. Then there was a deprotection–cyclization step to form two biological-activity-enticing imidazoquinoxaline families (Scheme 16) [42].

In 2016, the Pirali’s group reported a method for synthesizing 4-PI analogues via the van Leusen MCR, which is by far the most direct method for obtaining functionally rich imidazole. This transformation can be used to synthesize four different series of compounds—1,4,5-trisubstituted and 1,5-, 1,4- and 4,5-disubstituted imidazoles **71**. Meanwhile, 4,5-diaryl imidazoles were analyzed by a 3D quantitative structure-activity relationship (SAR). Based on their docking score and synthetic feasibility, the compounds were selected, synthesized, and biologically evaluated. Compared with 4-PI, the IDO1 inhibitor products through this experimental method have enhanced the potency. Both in enzymatic and cellular assays, the most active compounds showed lower micromolar potency, but no detectable cellular toxicity. The analysis displayed that a putative hydrogen bond between the Ser167 of the protein and the NH of the imidazole ring might be responsible for the improvement of the potency, together with favorable interactions with the partially occupied pocket B. The results above suggested that the 4,5-disubstituted imidazole framework might give a new orientation for the design of IDO1 inhibitors, in future (Scheme 17) [43].

At the same year, the Suresh group developed a direct, sequential, copper-catalyzed *N*-arylation–condensation reaction utilizing chiral cyclic 1,2-diamines **73** and ortho-haloaryl aldehydes or ketones **72**. The corresponding chiral tricyclic 1,4-benzodiazepines **74** were synthesized in high yield. Subsequently, 1,4-diazapine **75** was converted into a novel tetracyclic *N*-fused ImBDs through van Lusen imidazole synthesis (Scheme 18) [44].

A range of 5-aryl-1-alkylimidazole derivantes **79** were synthetized by utilizing the van Leusen MCR by Bojarski and co-workers, in 2017. The van Leusen reaction was performed by the stepwise cycloaddition of TosMIC **18** to the polarization double bond of a preformed imine **78**. Imidazole is synthetized by the elimination of *p*-TosOH, from the cyclic intermediate. Then, aromatic aldehydes bind to the appropriate amine. All compounds were highly selective to 5-HT5A. These compounds were metabolically stable in human liver microsome, showed less toxicity in HEK-293 and HepG2 cells and were water-soluble (Scheme 19) [45].

In 2019, they also synthesized a range of fluorinated indole-imidazoles to find more effective 5-HT7 receptor selective agonists as molecular probes. As shown in Scheme 20, the target compounds **82** were obtained from the relevant indoles undergoing the formylation process and the stepwise van Leusen imidazole synthesis with TosMIC **18**. Meanwhile, they indicated that compound **82a** might be a potential analgesic or a long-sought tool for studying the receptor effect of 5-HT7, based on the anti-nociceptive function on a murine model of neuropathic pain (Scheme 21) [46].

In 2017, Sharma and co-workers developed a brief, novel, and simple operational van Leusen method. They presented the first van Leusen process towards synthesis of high functionalized dihydrodibenzo[b,f]imidazo[1,2-d] [1,4] thia/oxazepine **86**, by reacting TosMIC **18** and dibenzo[b,f] [1,4] thia/oxazepines **85**, which were obtained from compound **83** and **84**, under a basic condition (Scheme 22) [47].

In 2018, the Weaver group reported that two corresponding series of molecules based on the SAR design—*N*1-substituted 5-indoleimidazoles and *N*1-substituted 5-phenylimidazoles were accessed. The latter (and more efficient) series were obtained by an accidental rearrangement of imines that are intermediate in the van Leusen imidazole synthesis reaction. The one-pot synthesis of compounds **89** in two steps utilizing the van Leusen imidazole synthesis process is shown in Scheme 23. First, imine formation between 2-indolecarboxaldeyde **87** and the appropriate benzyl amine **88** took place, which then reacted with TosMIC **18** and K_2_CO_3_ to give indoleimidazoles. They have also conducted SAR studies on their novel and more potent compounds. Compounds **92** were synthesized starting from known amine **91** and aromatic aldehyde **90** (Scheme 24). These inhibitors represent a promising future for the development of IDO1 inhibitors with specific physical and chemical properties that are easy to synthesize [48].

In the same year, Kondaparla et al. synthetized a range of short chain 4-aminoquinoline-imidazole derivatives **95** using two steps, in one-pot, via the van Leusen MCR standard method. The detailed synthetic protocol for the synthesis of target compounds **95** is shown in Scheme 25, compound **94** as amine was utilized to react with TosMIC **18** and acetaldehyde **93** through a multicomponent cyclisation reaction. In the end, all synthesized compounds were screened against the chloroquine sensitive and chloroquine-resistant strains of *Plasmodium falciparum*. In this study, substitution on the *N*-(2-(1*H*-imidazol-1-yl)ethyl)-7-chloroquinolin-4-amine moiety had greatly influenced the antiplasmodial bioactivity [49].

In the same year, Laali and co-workers reported that an available route provided access to various C5-substituted imidazoles **97**, **98**, and **99**, through one-pot tandem via the van Leusen-Suzuki, van Leusen-Heck, and van Leusen-Sonogashira methods, respectively. Imidazolium-ILs as solvents, along with piperidine-appended imidazolium [PAIM][NTf_2_] as task-specific basic IL and facile aldimines 96, and TosMIC **18** in the mild condition, were employed in their article (Scheme 26) [50].

In the late 2018, Guan and co-workers developed a novel and highly efficient synthesis of polysubstituted 1*H*-imidazo-[4,5-*c*] quinoline derivatives through the stepwise van Leusen/Staudinger/aza-Wittig/carbodiimide-mediated cyclization method. Azides **102** were accessed by the van Leusen reaction of 2-azidobenzaldehyde **100**, amine **101**, and TosMIC **18**. Then, 1*H*-imidazole-[4,5-*c*] quinoline **103** was formed by a tandem aza-Wittig reaction with isocyanate, in a moderate to good yield (Scheme 27) [51].

In 2019, Lammi et al. reported an imidazolyl peptidomimetic, which has shown proprotein convertase subtilisin/Kexin 9 (PCSK9) inhibitor bioactivity in the micromolar dosage range. As shown in Scheme 28, the first imidazole derivative **105** was accessed in good yield, starting from *p*-anisaldehyde **104**, methylamine **77**, and TosMIC **18**. Then, compound **105** was treated with *n*-butyllithium and DMF as a formylating reagent at low temperatures, to obtain the aldehyde derivative 106 in high yield. The target compounds 107a RIm13 and 107b RIm14 were accessed via a quasi-iterative protocol, with the emphasis on the alternating van Leusen three-component reactions with two formylation steps. In the end, they found that RIm13 represents currently one of the most potent proprotein convertase subtilisin/kexin 9/low-density lipoprotein receptor, (PCSK9/LDLR) protein−protein interaction of small molecules (Scheme 28) [52].

In 2020, Rashamuse and co-workers described a microwave-assisted cycloaddition of TosMIC **18** with imines and aldehydes to form 1-substituted 5-aryl-1*H*-imidazoles. Imidazoles **110** were also obtained in a one-pot, two-steps reaction with a yield comparable to that obtained through step-by-step irradiation of aldehyde **108** and aliphatic amine **109**, with a neat microwave at 60 °C for 4 min, followed by the addition of TosMIC **18**, K_2_CO_3_, and CH_3_CN, and the reaction mixture was placed under microwave conditions (Scheme 29) [53].

Moreover, the antibacterial properties of these fragments in vitro were tested by the minimum inhibitory concentration (MIC) bioassay. The results showed that the MIC value of compound **110a** against *Staphylococcus aureus* was 15.6 μg/mL, while **110b** displayed a similar MIC value against *Bacillus cereus*, and indicated that these compounds might be further developed to specifically target microbial pathogens (Scheme 30).

In 2014, the Bunev group reported a novel process for the synthesis of 1,4,5-trisubstituted imidazole-containing trifluoromethyl group **112**, which contained two-component condensation reaction, *N*-aryltrifluoroacetimidoyl chlorides **111** reacted with TosMIC **18**, as well as sodium hydride in dry THF at room temperature, under argon atmosphere (Scheme 31, top) [54].

Then, in 2019, they also previously described a procedure, in which 1-imidoylbenzotriazoles [*N*-aryl-1-(1*H*-benzotriazol-1-yl)-2,2,2-trifluoroethan-1-imines] **113** reacted with TosMIC **18**, according to the van Leusen reaction to obtain a good yield of 1-aryl-4-(4-methylbenzenesulfonyl)-5-(trifluoromethyl)-1*H*-imidazoles **114**, which is difficult to access. The yield of **114** almost did not depend on the substituent in the *N*-aryl fragment of initial imidoylbenzotriazole **113** (Scheme 31, bottom) [55].

A possible mechanism for synthesis of the imidazole-containing trifluoromethyl group is shown in Scheme 32. Initially, deprotonation of TosMIC with sodium hydride forms stabilized carbanion **18**, which attacks the carbon–nitrogen bond’s carbon atom of **115**, to give the intermediate adduct **116**. Elimination of the R ion from the latter generates intermediate **117**, which undergoes intramolecular cyclization and leads to imidazole **118**.

## 4. Other van Leusen Imidazole Synthesis

In 2015, Fodili and collaborators described the synthesis of a 1,4-disubstituted 5-methylimidazole **121**. As shown in Scheme 33, compound **121** was prepared by reacting enamine **119** with TosMIC **18**, under the presence of tert-butylamine and a catalytic amount of bismuth (III) triflate in methanol. In this research, it was the first example of a usual rearrangement in the van Leusen imidazole synthesis and showed that the imidazole ring system can be prepared through reaction with TosMIC and a tautomeric enamine, to form a secondary ketamine. The possible mechanism involves the formation of the van Leusen imidazoline intermediate, followed by a C–C bond cleavage and then subsequent tosyl substitution [56].

In 2019, the Suresh group demonstrated the formation of imidazoles in the presence of water as a solvent and a base-free condition. The reaction of dihydro *β*-carboline imines **122** and *p*-toluenesulfonylmethyl isocyanides **18** formed the corresponding substituted *N*-fused imidazo 6,11-dihydro *β*-carboline derivatives **123**, with good yields under mild and green condition (Scheme 34) [57].

A possible mechanism for the present metal- and base-free imidazole framework is shown in Scheme 35. Initially, the precursor dihydro *β*-carboline imine **122** acts as a base that captures proton from TosMIC **18**, to provide a C-nucleophile, which would add to the dihydro *β*-carboline imine **122**, and then be cyclized to form the intermediate **125** through the intermediate **124.** Another molecule of the starting dihydro *β*-carboline imine **122** captures a proton from intermediate **125**, then removes the tosyl group, which might result in the construction of imidazole derivative **123**. Subsequently, the product imidazole **123** might also act as a base following the rational reaction mechanism, as described in Scheme 35.

At the same year, Necardo et al. found an unusual multicomponent synthesis of 4-tosyl-1-arylimidazoles **127**, by considering aryl azides as the electrophilic partners for the TosMIC **18**-mediated van Leusen cycloaddition. In this transformation, it is the first example of the reaction of two TosMIC molecules participating in van Leusen imidazole synthesis (Scheme 36) [58].

A plausible scenario for the MCR is shown in Scheme 37. In the initiation step, the TosMIC anion attacks *N*-3 of the azide **128** to produce intermediate **129**. Then, *N*-1 intercepts the isocyanide in a 6-endo-trig cyclization to form anion **130**, which is quenched by a proton source to give **131**. Owing to its instability, compound **131** processes a [4 + 2] cycloreversion to formation **132**, with a loss of nitrogen. Subsequently, the imine of **132** passes through an attack by a second molecule of the TosMIC anion, followed by ring closure, to produce **133**. At this point, after protonation, intermediate **134** regains aromaticity via a base-assisted mechanism, with the expulsion of the most acidic proton and loss of hydrogen cyanide and sulfinate. Under strong basic condition, excess *t*-BuOK deprotonates the newly formed hydrogen cyanide, avoiding the release of toxic HCN. It is worth noting that the two molecules of TosMIC partake in the reaction mechanism in two different paths, with the second molecule in Scheme 37 undergoing a fragmentation resulting in the incorporation of a C−H into the target molecule.

In 2019, the one-pot three-component van Leusen chemistry was used for the DNA-encoded libraries (DELs) synthesis by the Staz group, which provided the first published DNA-compatible approach to form unusual highly functionalized imidazoles. The target DEL productions **138** were obtained from the aldehyde functionalized DNA molecule **136**, amine **137**, and TosMIC derivatives **18** by the van Leusen reaction. Moreover, a wide variety of amines, commercial TosMIC molecules and aldehyde as the variety element in the three ingredients heterocyclization were investigated under an optimized condition, respectively. This transformation meaningfully expanded the application of van Leusen imidazole synthesis and DNA compatible chemistries (Scheme 38) [59].

## 5. Conclusions

In summary, under the in-depth research and application in imidazole-based medicinal chemistry and the progress in other disciplines—such as cell biology, molecular biology, pharmacology, and organic chemistry—an increasing number of imidazole-containing drugs with lower toxic, better efficacy, superior pharmacokinetic characteristics, effective pathologic probes and diagnostic agents would be used in clinics. This could make remarkable contributions for the protection of mankind’s health. Therefore, the van Leusen imidazole synthesis based on TosMICs will play an increasingly central part in the synthesis of bioactive compounds as clinic drugs in drug design and synthesis. We could focus on the changing of the various aldimine groups and TosMIC derivatives in the van Leusen reaction to modify the imidazole derivatives in future. Above all these have clearly indicated the infinite potentiality of van Leusen imidazole synthesis in medicinal chemistry. Additionally, we hope this review would build a full foundation and reference source which would open up new thoughts for researchers to focus on in imidazole-based medicinal molecule design and synthesis chemistry.

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
