# Peer review of "Synthesis of Imidazole-Based Medicinal Molecules Utilizing the van Leusen Imidazole Synthesis"

_pharmaceuticals, 2020, doi:10.3390/ph13030037_

Round 1

Reviewer 1 Report

Imidazole derivatives are important heterocyclic compounds for medicinal chemistry and pharmaceutical science. In this context, the review topic is actual for a broad range of researchers from both academia and industry.

However, the review contains just 60 references, which too little (in my opinion) for this kind of papers. Moreover, most of these references are published before the 2015 year. I have a serious doubt, that this review will be useful for readers. The authors should update their manuscript with a lot of fresh references.

Author Response

Dear reviewer,

We are very glad to receive your review regarding our manuscript “Synthesis of imidazole-based medicinal molecules utilizing the van Leusen imidazole synthesis” (pharmaceuticals-728900).

Thank you for the helpful suggestions. According to your comments, we revised the manuscript carefully, the language was polished and the modifications are highlighted in green.

Please find the response to the comments in the following annex.

Thanks again for you to review this manuscript.

Sincerely yours,

Dr. Dawei Zhang

College of Chemistry

Jilin University

2699 Qianjin Street

Changchun, 130012, P. R. China. 

Tel (Fax) +86-431-8783-6471

Response to reviewer 1 comments

Question:

Imidazole derivatives are important heterocyclic compounds for medicinal chemistry and pharmaceutical science. In this context, the review topic is actual for a broad range of researchers from both academia and industry.

However, the review contains just 60 references, which too little (in my opinion) for this kind of papers. Moreover, most of these references are published before the 2015 year. I have a serious doubt, that this review will be useful for readers. The authors should update their manuscript with a lot of fresh references.

Answers:

The van Leusen reaction based on TosMICs is one of the most important strategies to formation of five-member heterocyclic compounds such as pyrroles, imidazoles, oxazoles, etc. Due to its excellent advantages, the van Leusen reaction has been developed rapidly into van Leusen pyrrole synthesis, van Leusen imidazole synthesis, van Leusen oxazole synthesis and others in the past decades and many articles have been reported every year.

Because we have previously published a review about van Leusen reaction for the preparation of pyrrole heterocycle in Molecules (2018, 23, 2666), this review is focus on the formation of imidazole heterocycle via van Leusen imidazole synthesis and as an important part of van Leusen reaction.

Indeed, it is less references after 2015 year, but the van Leusen imidazole synthesis might be rapidly developed based on the importance of imidazole-containing medicinal molecules in future. Therefore, we finish this manuscript and expect that it is advantageous to promote the application of this reaction in medical heterocyclic chemistry.

Thanks again for your comments.

Reviewer 2 Report

The authors describe medical molecules based on the imidazole ring. The work is very interesting and will interest many researchers involved in the design and synthesis of substances with desired biological properties. However, I have a few comments and opinions that the authors should comment on:
1. Scheme 3 (mechanism). On what basis do the authors propose such a scheme of synthesis? Have such tests been done before?
If so, there is no reference to literature. If there were no such studies, they shuld be the subject of another publication. In general, schemes relating to reaction mechanisms are unnecessary in this study because they disturb the uniformity of work. In addition, they do not bring relevant information in the context of medical chemistry.

2. Literature should be reviewed and standardized once again. There are many mistakes especially in the abbreviations of the cited magazines.

3.In addition, the Schemes should be standardized. The methyl group is sometimes as CH3 sometimes as Me for example Scheme 8, Scheme 9, Scheme 14, Scheme 30.

4. Also, why 6-nitropiperone is capitalized (line 97).

Author Response

Dear reviewer,

We are very glad to receive your review regarding our manuscript “Synthesis of imidazole-based medicinal molecules utilizing the van Leusen imidazole synthesis” (pharmaceuticals-728900).

Thank you for the helpful suggestions. According to your comments, we revised the manuscript carefully, the language was polished and the modifications are highlighted in green.

Please find the response to the comments in the following annex.

Thanks again for you to review this manuscript.

Sincerely yours,

Dr. Dawei Zhang

College of Chemistry

Jilin University

2699 Qianjin Street

Changchun, 130012, P. R. China. 

Tel (Fax) +86-431-8783-6471

Response to reviewer 2 comments

Questions 1:

Scheme 3 (mechanism). On what basis do the authors propose such a scheme of synthesis? Have such tests been done before?
If so, there is no reference to literature. If there were no such studies, they should be the subject of another publication. In general, schemes relating to reaction mechanisms are unnecessary in this study because they disturb the uniformity of work. In addition, they do not bring relevant information in the context of medical chemistry.

Answers 1:

The Scheme 3 (mechanism) was described based on the reference 22, and the mechanism was described detailedly in this article.

This reference is important and the first example of van Leusen imidazole synthesis. We believe that it is necessary to display this mechanism and more easily to understand this reaction for readers.

Questions 2:

Literature should be reviewed and standardized once again. There are many mistakes especially in the abbreviations of the cited magazines.

Answers 2:

Some mistakes have been made carefully corrections. The modifications are highlighted in green.

See line 403-552, please.

Questions 3:

In addition, the Schemes should be standardized. The methyl group is sometimes as CH3 sometimes as Me for example Scheme 8, Scheme 9, Scheme 14, Scheme 30.

Answers 3:

These questions of Scheme have been annotated and made corrections under maintaining the primary framework in this manuscript.

See the revised manuscript, please.

Questions 4:

Also, why 6-nitropiperone is capitalized (line 97).

Answers 4:

This mistake has been made corrections.

See line 101 in revised manuscript, please.

Thanks again for your comments.

Round 2

Reviewer 1 Report

I give my confirmation for publishing this manuscript in the present form.